# Education/Training Interventions to Improve Oral Health of Adults with Intellectual Disability

**DOI:** 10.3390/healthcare10102061

**Published:** 2022-10-17

**Authors:** Tejashree Kangutkar, Hanny Calache, Joanne Watson, Susan Taylor, Susan Balandin

**Affiliations:** 1Rural Health School, La Trobe University, Edwards Road, Bendigo, VIC 3550, Australia; 2School of Health & Social Development, Deakin University, 221 Burwood Highway, Sydney, VIC 3125, Australia

**Keywords:** oral health training, people with intellectual disability, dental health professionals, primary carers

## Abstract

People with intellectual disability have poorer oral health outcomes and experience greater difficulties accessing dental services than the rest of the community. Interdisciplinary educational programs for those involved in oral health decision making for people with intellectual disability aim to improve their oral health. This scoping review explored education/training interventions related to the oral health of adults with intellectual disability. Interventions targeted dental health and non-dental health professionals, adults with intellectual disability, and their paid and unpaid supporters. Six electronic databases were searched using PRISMA guidelines and the Crowes critical appraisal tool. The search strategy, plus the application of the inclusion and exclusion criteria, presented in the body of the manuscript, led to the selection of 20 papers for inclusion into this review. The quality of undergraduate tuition relating to disability was reported to be poor, but students and dental health professionals expressed interest in expanding their knowledge in this area. Disability-support workers and primary carers of people with intellectual disability reported improved knowledge and attitude, but the oral health status of the people they supported was not impacted. No research was found that explored the perspectives of non-dental health professionals. Oral health training/education contributes to systemic barriers in general health, wellbeing, and social inclusion of people with intellectual disability. Revision of disability-related programs in general dentistry courses and longitudinal impact research is needed.

## 1. Introduction

Adults with intellectual disability as a group have poorer oral health than the general population [1]. This in turn impacts their quality of life [2,3] and overall general health. The impact of oral health or diseases on an individual’s daily functioning, well-being, or overall quality of life [4] is reflected in models of Oral Health Quality of Life (OHQoL) that recognise five consequences of oral disease: “impairment, functional limitation, pain/discomfort, disability, and handicap” [4,5]. Poor oral health is strongly associated with chronic disease such as diabetes, cancer, cardiovascular disease, respiratory disease, and stroke [6] as well as conditions highly prevalent in the intellectual disability population, such as aspiration pneumonia and gastroesophageal reflux disease (GERD) [2,3]. Oral-health-related pain can impact nutritional intake along with a person’s ability to perform daily tasks. In addition, halitosis because of poor oral health can impact the ability of a person with intellectual disability to interact socially, contributing to poor psychological and social health. Current evidence has demonstrated a link between oral diseases (dental caries, periodontitis, oral mucosal lesions) and chronic conditions such as diabetes, cardiovascular diseases, and more recently, cognitive impairment [7,8,9].

Over one billion people globally or 15% of the world’s population experience disability [10]. Physical constraints, poor financial status, difficulties related to prevailing health conditions, inadequate availability of trained carers, and stigmatization are some of the many barriers this population faces when accessing healthcare services [10]. This is despite Article 25 of the United Nations Convention on the Rights of Persons with Disabilities (UNCRPD) promoting equal access to health care services [11]. The UNCRPD states that parties shall:

“Provide those health services needed by persons with disabilities specifically because of their disabilities, including early identification and intervention as appropriate, and services designed to minimize and prevent further disabilities, including among children and older persons” [11].

People with disability, including those with intellectual disability, have poorer health outcomes than the general population [10,12]. In most developed countries, including Australia, public dental services are available to all school children up to age 18. However, for adults, access to public dental services is much more restricted [13]. Public dental services in Australia provide emergency and general dental and denture care for adult concession cardholders (i.e., low income). A small fee is payable at each visit up to a set maximum. Advanced dental treatments may be provided at fees above the maximum for groups that meet the eligibility criteria.

Existing evidence suggests that people with disability have poorer oral health than the general population of people with unmet oral health needs [14]. Poor oral health has both physical and social consequences for people with intellectual disability, including increased morbidity and social exclusion [6,15], as reflected in OHQoL models [4].

Education/training of health professionals aids early identification and prevention of further serious health issues for adults with intellectual disability. It creates increased awareness and skills and encourages health service providers to adopt positive attitudes and confidence towards improving the well-being of people with disability [10].

The first aim of this scoping review is to explore the evidence relating to educational interventions targeting dental health professionals, non-dental health professionals, people with intellectual disability, and their support workers and primary carers with the intention of improving the oral health of people with intellectual disability. The second aim is to identify evidence gaps and provide recommendations on improving the oral health and OHRQoL of people with intellectual disability.

The objectives of this review are to understand:Oral health education/training interventions for adults with intellectual disability that target dental health professionals. (Dental health professionals diagnose and treat diseases of the teeth, gums, and mouth, including the prevention of dental diseases. They include dentists, dental hygienists, dental therapists, dental prosthetists, and oral health therapists);Education/training programs about the importance of oral health for adults with intellectual disability, which target non-dental health professionals who support people with intellectual disability. (Non-dental health professionals include general medical practitioners, nurses, diabetes educators, speech pathologists, pharmacists, midwives, etc. Non-dental health professionals could provide oral health promotion, temporary relief of oral symptoms and referral rather than definitive treatment due to lack of oral health training and scope of practice limitations);Education/training programs about the importance of oral health that target adults with intellectual disability;Education/training programs relating to oral health interventions that target primary carers and support workers of adults with intellectual disability.

## 2. Materials and Methods

This scoping review follows the Preferred Reporting Items for Systematic Reviews and Meta-Analyses (PRISMA) 2020 checklist [16]. The objective of the scoping review is to thematically synthesise the evidence of educational interventions targeting dental health professionals and non-dental health professionals, to identify evidence gaps and recommend strategies to improve the oral health and OHRQoL of people with intellectual disability.

### 2.1. Inclusion Criteria

The search was limited to full-text, peer-reviewed academic journal articles written in English and published between years 2008 to 2019. Research was included if it focused on: adults aged over 18 years with intellectual disability, highlighted education, and training as the prime area of focus, and explored issues related to the understanding of oral health needs of adults with intellectual disability.

### 2.2. Exclusion Criteria

Studies excluded from the review were those that were not published in English and in peer-reviewed journals; studies exploring disabilities other than intellectual disability (e.g., physical disability or ageing); studies exploring strategies other than education and training; studies exploring the prevalence, aetiology, risk predictors, or clinical treatment of oral health issues for people with intellectual disability; studies that targeted education/training interventions for population other than adults with intellectual disability; and published research conducted prior to the year 2008, in consideration of changing oral health practices.

### 2.3. Literature Search

Relevant articles were identified by means of an electronic search using the *EBSCO Host* platform to access *Academic Search Complete, CINAHL Complete, Global Health, Health Source Nursing/Academic Edition, MEDLINE Complete*, and *MEDLINE* from 2008 to 2019. In addition, *Google Scholar* was searched, and the first author hand-searched reference lists of eligible articles to maintain rigour. Articles for each of the four objectives described above were searched separately using appropriate search terms with Boolean operators “OR” and “AND” to locate relevant records from the databases (Table 1). The ‘asterisk’ symbol used in the key search terms (Table 1) is known as a truncation symbol. The asterisk * symbol enables the search for different word endings. For example, **teen*** will identify results for the following words: teen, teens, teenager, teenagers.

### 2.4. Study Selection

Figure 1 below sets out the literature search and selection process. The initial database search revealed a total of 736 published articles. The 736 published papers related to study objectives as follows:

Objective 1: Oral health education/training interventions for adults with intellectual disability that target dental health professionals: 270 published articles;

Objective 2: Education/training programs about the importance of oral health for adults with intellectual disability, which target non-dental health professionals who support people with intellectual disability: 88 published articles;

Objective 3: Education/training programs about the importance of oral health that target adults with intellectual disability: 341 published articles;

Objective 4: Education/training programs relating to oral health interventions that target primary carers and support workers of adults with intellectual disability: 37 published articles.

A total of 557 articles were excluded because they were not published in peer-reviewed journals, or their focus was not on the study objectives. Results were merged using Endnote X8™ reference-management software, and duplicates were removed. The first author hand-searched the reference lists to identify any further relevant published research.

The titles and abstracts of the resulting 179 articles were independently screened by two reviewers independent of the research, resulting in 40 articles eligible for full-text screening. Following the screening of the full texts, 20 articles were deemed eligible to be included in the review. Articles excluded either were not related to education/training-based interventions, or the population being studied were children rather than adults. Eligibility of articles for final inclusion was discussed independently between two reviewers until consensus was reached. The Cochrane Collaboration [17] recommends that more than one individual independently screens and selects studies to minimize bias and human error and to help ensure that the selection process is reproducible. Of the 20 studies that were found to meet the inclusion criteria, 17 used quantitative methods, 3 used qualitative methods, 1 was a cost-effectiveness analysis, and 1 was a systematic review.

Of the twenty included studies, twelve addressed objective 1, none addressed objective 2, two studies addressed objective 3, and six studies addressed objective 4.

Relevant findings were extracted from the eligible papers and exported to a Microsoft Excel 2010 spreadsheet by the first author. The data included quotes from each paper grouped together to form key themes. Themes were integrated by discovering common trends in key research findings. Meta-analysis was not conducted due to the nature of objectives and variability in study outcomes. A narrative approach was used for reporting the findings of all eligible studies irrespective of their study design. Multiple study designs were found, including synthesis (systematic reviews), exploratory, observational (cross-sectional, survey), and experimental studies. The first author extracted the data from eligible studies to a standardised format, and these data were verified by the other two investigators.

## 3. Results

Table 2 summarises each study type, its relationship to study objectives and populations, sample sizes, and research findings. Studies found were conducted in France, Italy, Japan, Ireland, Iran, the USA, the UK, Poland, Australia, Belgium, and the Netherlands. Thirteen out of twenty studies were published between 2015 to 2018. No study included in this review was published before 2010, demonstrating an increased interest in this topic during recent years.

### 3.1. Characteristics of Participants

Research participants included dental health professionals, people with disability, and carers of people with disability. No study was found that included research into education/training/experiences of non-dental health professionals (objective 2 of this literature review).

### 3.2. Interventions

This review focused on interventions related to education/training in oral health of adults with intellectual disability. Interventions developed training modules or evaluated effectiveness of training modules and were evaluated by measuring impacts upon knowledge, skills, and attitudes. Only one study focused on the development of scales or assessment tools to measure these variables.

### 3.3. Synthesis of Results

#### 3.3.1. Objective 1: Education/Training-Related Interventions for Dental Health Professionals

Eleven of the twenty published studies reported on education/training of dental health professionals and their experience and expectations in managing the oral health of people with intellectual disability [18,19,20,21,22,23,24,25,26,27,28]. Research was conducted, variously, with undergraduate and senior dental students, practising dentists, and dental hygiene students. All 11 studies used quantitative research methods, and all were published within the past decade.

Below, we categorise the 11 studies as either experimental or exploratory. Four experimental studies evaluated the effectiveness of education/training interventions. Eight exploratory studies examined either (a) assessment of the knowledge, skills, and attitudes of dental health professionals towards people with disability or (b) scales or tools with which to assess such knowledge, skills, and attitudes.

#### 3.3.2. Category 1: Experimental Studies

Researchers used experimental studies to examine training/education delivered to practising dentists and dental students. Research focused on two types of learning modules: online learning and modules delivered by DVD.

Catteau et al. [19] indicated that online training modules on dental care for people with disability were effective when delivered to 24 practising dentists and argued that this approach may therefore have value for undergraduate and postgraduate dental education. In the case of Mac Giolla Phadraig, Nunn [29], the intervention consisted of a special-care dentistry module of lectures and experiential workshops, e-learning, online tutorials, and peer observation during the third term of a three-year dental training trialled over a three-year period at undergraduate level. Analysis using attitudinal scores found the module failed to produce any statistically significant difference in students’ attitudes toward people with disability [29]. However, the researchers identified shortcomings in the assessment instrument used and proposed new measures were needed to measure changes in attitudes in dental students. Mohebbi, Chinipardaz [24] used Likert scale analysis of pre- and post-surveys to suggest that, although a short-term training module can have a positive impact upon students’ knowledge and attitude toward people with disability, further longitudinal research into knowledge retention is needed.

Educational modules delivered by DVD featured the lived experience of people with disability of oral health management and treatment and included self-reflections and student discussions in research by Jones and Miller [25]. Data from questionnaires indicated modules had positive impacts upon attitude and empathy toward people with intellectual disability on the part of dental hygiene students. Similar to Mohebbi, Chinipardaz [24], the researchers recommended further longitudinal research.

#### 3.3.3. Category 2: Exploratory Studies

Nine exploratory studies focused on assessment of knowledge, skills, and attitudes of dental health professionals toward people with disability. Six studies were used to explore such impacts [18,20,22,26,27,30]. Unlike other research in this review, a Delphi study by Blaizot, Catteau [30] with special needs dentists (SNDs) as panellists noted the co-operative nature of dentistry, bringing together treatment by the dental practitioner and the maintenance of dental hygiene at home. Some treatments were easier than others to discuss with patients or their carers, leaving open the possibility that shared decision making may not always occur. The authors called for cultural change amongst dentists, general practitioners, organisations, patients, and caregivers in the interests of relational autonomy. Shared attention to “the relational and emotional dimensions of care” [30] could evolve into a team approach to the oral health of this group. The authors noted the development of an undergraduate and postgraduate curriculum by the International Association for Disability and Oral Health, learning outcomes that included critical thinking about SND and positive attitudes toward people with disability.

Low levels of training in treatment of people with disability were noted by Dellavia, Allievi [20]; Smith, Rooney [27]; and Ahmad, Razak [18]. Survey response rates were low in the first two of these studies (51% and 35%, respectively), but evidence from these three studies suggested that clinical training is needed at undergraduate and postgraduate levels and that postgraduate students did not feel comfortable treating people with intellectual disability [18]. This reported discomfort may support findings by Smith, Rooney [27] that the management of dental emergencies was the most common service delivered to this group of patients.

The benefits for people with intellectual disability of treatment by experienced dental health professionals were evident in research by Nagarajappa, Tak [26] and by Isobe, Izumi [22]. Nagarajappa, Tak [26] reported that dental health professionals’ past experience of treating people with learning difficulties and their higher qualifications were associated with positive attitudes towards people with learning disability. Isobe, Izumi [22] similarly found attitudes of dental hygienists toward adults with dementia were more positive than those of dental hygiene students [22].

Only one study was found [21] on any scales or tools with which knowledge, skills, and attitudes toward oral health for people with intellectual disability may be measured. The study concluded that a research battery developed by researchers was effective in measuring values, attitudes, and intentions of dental health undergraduate students toward people with intellectual disability. The researchers intended to follow this research with field validation [21].

#### 3.3.4. Objective 2: Education/Training-Related Interventions for Non-Dental Professionals

As explained above in the description of method, no published papers were found that reported on oral health education/training interventions for non-dental health professionals. We note that research by Blaizot, Catteau [30], discussed below under objective 3, referred to other professionals who may support people who have limitations with their decision making, such as speech and language therapists, occupational therapists, and educational support. However, this was not developed as a theme by the researchers.

#### 3.3.5. Objective 3: Education/Training-Related Interventions for People with Intellectual Disability

Of the 20 published studies, only one exploratory study by Lees, Poole [31] examined education/training for people with intellectual disability. Also included in this objective 3 is research by Mac Giolla Phadraig, Dougall [32], who sought the perceptions of a small group of people with learning disabilities regarding how dental services could best meet their needs.

The small study by Lees, Poole [31] aimed to explore the experiences and expectations of six carers and four adults with intellectual disability who used community dental services. The authors found that carers were satisfied with the skills and competence of dental health professionals who treated people in their care. Positive themes were practices that respected the dignity of people with intellectual disability, such as good interpersonal communication, providing information, and actively listening. Negative factors were high dissatisfaction with transition from child to adult dental services and the anxiety-provoking effect of waiting times for patients with intellectual disability. The researchers recommended further research with a wider population of people with intellectual disability but proposed that promoting self-care in a format appropriate for people with intellectual disability would encourage independence in emergency and out-of-hours situations.

Conversely, Mac Giolla Phadraig, Dougall [32] found six focus group participants with learning disabilities were disempowered when they received dental services. Predominant themes included lack of information, lack of timely treatment of dental problems, minimising time in dentists’ waiting rooms, choice of dental services, the quality of care, and cost. The authors proposed measures for the empowerment of people with learning disabilities, such as inclusive service planning and appraisal and comprehensive, accessible information about dental services.

#### 3.3.6. Objective 4: Education/Training-Related Interventions for Carers of People with Intellectual Disability

Six of the twenty published papers in this review addressed education/training interventions for carers of people with intellectual disability. In an exploratory study, Eijsink, Schipper [33] developed a methodology to identify the prevailing viewpoints of carers in order to inform the development of a targeted education/training intervention. Q-methodology was used to collect data on caregivers’ attitudes to oral health care, which were categorised as responsible and perseverant, motivated, and aware of obstacles, social-minded and knowledgeable, and concerned and insecure. The researchers argued that being able to categorise caregivers’ attitudes toward oral health would help dental health professionals to personalise their advice and to better understand obstacles to the oral health of people with intellectual disability.

Four experimental studies examined the impacts of education/training interventions for carers [23,34,35,36]. Pradhan et al. (2016) measured the knowledge, confidence, and skills and dental efficacy of 41 participant carers who had received education or training in oral health for 103 adults with intellectual disability. The research included oral presentations and practical sessions, provision of oral hygiene care, and assessing the need for dental referral. Pre- and post-training evaluation demonstrated that carers’ knowledge and confidence had improved, but there was no evidence of significant increase in skills. The authors concluded that further research is needed with a larger cohort of carers and adults with intellectual disability.

Mac Giolla Phadraig, Nunn [35] conducted a randomised controlled trial (*n* = 50) to assess the impact of a multi-tiered oral health education programme on carers of people with intellectual disability. The intervention group received training on oral health and oral health diseases. A control group received no training. The training was measured for its impact on carers’ attitudes, self-efficacy, reported behaviour, and knowledge, but the researchers did not report on the effectiveness of the intervention on the oral health status of the people with intellectual disability receiving support from the carers. Whilst there was some limited improvement in carers’ knowledge of oral health, the research found no improvement in carers’ attitudes toward oral health of people with intellectual disability. The researchers later analysed the same data to determine any impact of carer training upon the dental health of participants with intellectual disability [23]. No significant improvement was found. A separate cost-effectiveness analysis [28] determined that, despite its small effect, the intervention was cost-effective, due to its low cost. The researchers proposed further cost–benefit analysis was required.

Similarly, Phlypo, Marks [36] found little impact on caregivers or disability care residents of a program of information, pre- and post-tests of dental health, and follow-up questionnaires of caregivers. The researchers suggested strategic guidelines would enhance learning at undergraduate level. A systematic review by Rozas, Sadowsky [37] reported on findings from nine studies. Roza and colleagues argued that dental health professionals played a vital role in educating caregivers and that the basic oral healthcare plan for people with dementia and cognitive impairment should match prevention strategies that are recommended for general elderly persons.

## 4. Discussion

This review sought evidence that the oral health of adults with intellectual disability can be improved by education and training interventions targeted at building knowledge, skills, and confidence of both the individual themselves and their support network. An individual’s support network may include disability support workers, primary carers, dental health professionals including SNDs, and non-dental health professionals. The research indicates that education and training targeted at building an understanding of the role oral health plays in the quality of life, health, and wellbeing of adults with intellectual disability has the potential to improve their oral health. Moreover, there is a need for education and training interventions designed to build the skills of people with intellectual disability and their supporters in oral health care. However, there is little evidence that current dental health training equips dental health professionals with the knowledge, attitudes, or skills needed to treat people with intellectual disability. Although one reviewed study adds to knowledge about good practice dental health care for people with intellectual disability [29], there is little empirical examination of practices such as referral of patients to SNDs, planning and preparation for dental treatment, and collaboration by medical practitioners in oral health care.

Well over half (12 of 20) of all the articles reviewed were relevant to objective 1 (to explore training interventions related to people with intellectual disability targeting oral health professionals). This is indicative of an increasing empirical interest in understanding the role of dental health professionals in improving the oral health of people with intellectual disability through training interventions. Evidence from these studies not only highlights a high level of student interest in pursuing SND after graduation but also suggests that the incorporation of education/training modules related to SND into undergraduate dental curriculum is likely to significantly improve knowledge and attitudes towards people with disability. Of note are the finding of Isobe and colleagues [22], who found that registered dental hygienists had more knowledge and positive attitudes towards people with dementia compared with dental hygiene students, pointing to a need for training in the treatment of people with dementia to be added to the dental hygienists curriculum. Although Isobe et al.’s study was focused on people with dementia as opposed to intellectual disability, its findings may also point to a need for increased training in the dental hygienists’ curriculum around other cognitive disabilities, including intellectual disability.

Despite this evidence, none of the studies reviewed found any causal relationship between these education/training interventions and improvement in the oral health status of people with intellectual disability. Moreover, although the incorporation of education/training modules relating to SND are likely to contribute positively to knowledge and attitudes, students generally rated the quality of education they received in the area at undergraduate level as poor.

On the other hand, no research was found to be relevant to objective 2 (to explore training interventions related to people with intellectual disability targeting non dental health professionals). However, the authors of this review argue that for an interdisciplinary approach to oral health to occur, it is imperative that all health professionals be educated in the importance of oral health for the management of the conditions that they are treating. Access is needed to both public and private dental services, and referral pathways are required for the smooth transitions of patients from one service to another.

Only two of the articles reviewed were considered relevant to objective 3 (to explore training interventions related to education/training for people with intellectual disability). These two studies explored the views of people with intellectual disability about their experiences with dental health services but did not report on the management of their oral health more generally. Both Mac Giolla Phadraig, Nunn [28] and Lees, Poole [30] identified a need for dental health professionals to better understand how to communicate verbally with people with intellectual disability and how to respect their dignity when managing their oral health. This suggests the principles of patient-centred care need to be introduced to dental health professionals through educational curricula and through patient advocacy.

Six of the articles reviewed were relevant to objective 4 (to explore interventions related to education/training for carers of people with intellectual disability). There is evidence in four studies [34,35,36,37] that there is a role for carers in maintaining the oral health of people with intellectual disability and that it is possible for oral-health-related education and training interventions for carers to improve their knowledge, attitude, self-efficacy, and behaviour towards oral health. For instance, the comprehensive approach taken by Pradhan, Keuskamp [34] involving dental professionals over a period of 3 months demonstrated substantial improvements in carers’ knowledge and confidence. However, other research demonstrated limited improvements in carers’ knowledge [36] and inconclusive evidence of improvements in attitudes across two studies by Mac Giolla Phadraig, Nunn [35] and [23]. As with dental health professionals, training interventions with carers led to no significant improvement in the oral health of care recipients. Analysis of cost effectiveness of a train-the-trainer approach to interventions with carers [28] found benefits relative to direct training of carers, suggesting that if an appropriate model was developed, its adoption could be beneficial for people with intellectual disabilities.

### 4.1. Implications for Research, Policy, and Practice

Further research is needed into the barriers preventing the improvement of the oral health of people with intellectual disability, with reference to models of care, surgical interventions compared with preventative care, the adequacy of training content, and issues in translating training content into dental health practice. Research is needed to discover what people with intellectual disability need to know about oral health and what dental health students need to know about people with intellectual disability. Holistic care approaches should be encouraged at the tertiary level.

Standardised education and training for dentists would promote patient-centred care as guided by the curriculum developed by the International Association for Disability and Oral Health and providing opportunities for students to be exposed to clinical settings outside the dental clinic, such as rehabilitation centres, residential aged care, and community centres. If integrated with transdisciplinary and interdisciplinary training and practice improvement for dentists, GPs and allied health, disability support professionals, and family/carers, this approach could facilitate a systemic shift in oral health training and professional practice. It is imperative that all health professionals be educated and trained in the importance of oral health in the management of the conditions that they are treating, the current evidence-based oral health preventive messages and procedures that are available, access to both public and private dental services, and the referral pathways required for the smooth transitions of patients from one service to another. Non-dental health professionals should be linked with dental practitioners and undertake interprofessional training to identify serious dental health issues that require immediate attention.

Although a number of studies included in this review [19,23,24,25,34] indicated that education/training for dental health students and for carers improved their knowledge and attitude toward the oral health of people with intellectual disability, researchers have not demonstrated positive impacts upon their oral health. Only Phlypo, Marks [36] proposed slight improvements in oral health. Further studies should focus on identifying barriers encountered by dental health professionals in implementing training content in practice. More research with larger sample sizes is required in testing effectiveness of interventions in improving oral health status of people with intellectual disability. Further development and use of assessment scales to evaluate values, attitudes, and intentions should be considered while developing education and training modules to understand views of dental health professionals towards SND and identify challenges from their perspective.

Training for carers of people with intellectual disability must primarily focus on prevention and oral hygiene care. Carers should also be trained to identify chronic dental problems and learn dental care protocols to follow during an emergency or out-of-hours situations. There are examples (e.g., Mac Giolla Phadraig, Guerin [23]) of modules designed for carers and other stakeholders to train people with intellectual disability on how to keep their teeth healthy. Despite their potential involvement in oral health care decision making for people with intellectual disability, the authors of this review found no published research that explored the perspectives of non-dental health professionals such as general medical practitioners and medical specialists in other fields.

### 4.2. Limitations

One of the negative implications of incorporating education and training modules in SND at undergraduate level to improve knowledge and skills is an already crowded curriculum with insufficient time to include any additional teaching. This is especially pertinent when a well-designed study with a large sample size [23] reported that such educational interventions had no significant impact on students’ attitudes and showed no significant improvement in oral health of their patients. Further research is recommended to evaluate the oral health status of people with intellectual disability after training among dental health professionals and carers. Studies should not be limited in focus to changes in knowledge and attitudes of students and dental health professionals but should also include oral health status and should be longitudinal in scope, as recommended by Mohebbi, Chinipardaz [24] and Jones and Miller [25]. It is also necessary to consider support worker and primary carer burden when incorporating oral health care into their routine.

A limitation of this review was inclusion of papers published in the English language only, which may have omitted important papers published in other languages.

## 5. Conclusions

Poor oral health affects systemic health and quality of life [38]. For people with intellectual disability, oral health is a factor in their general health and wellbeing and in their social inclusion. The research included in this review demonstrates that people with intellectual disability have experienced dental health care as disempowering and inaccessible, and no evidence was found of positive impacts of education/training of dental health professionals or carers on oral health status. Despite this, dental health students and professionals were interested in expanding their knowledge and practice skills in oral health care for people with intellectual disability. Although there is evidence that measures such as training carers in oral health have potential to improve referrals to SNDs, the interest of non-dental health professionals in the oral health of their patients with intellectual disability is currently unknown in published research. Further research is needed to identify systemic barriers to the oral health of people with intellectual disability and to progress the early work identified in this review towards developing effective education/training modules for dental health professionals, students, and carers that will improve their oral health.

## Figures and Tables

**Figure 1 healthcare-10-02061-f001:**
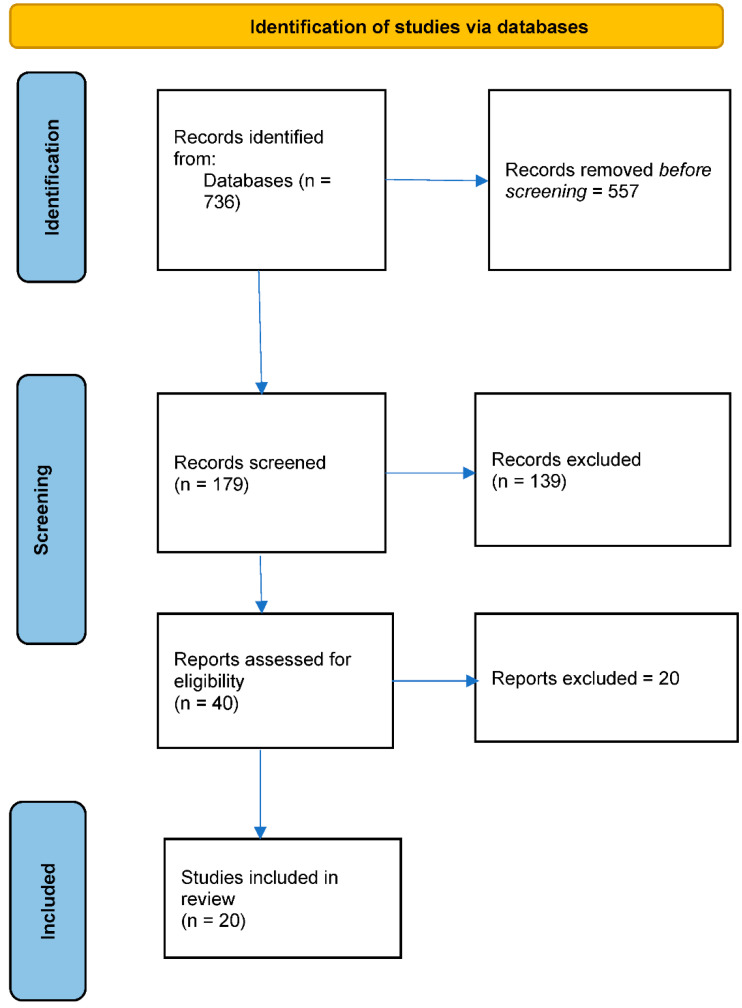
Identification of studies.

**Table 1 healthcare-10-02061-t001:** Summary of key search terms.

Objective 1			
Oral health related	People with disability related	Knowledge related	
“oral health” “oral hygiene” “dental health” “dental care” “oral care” “mouth care”	“people with disabilit*” “disabled people” “special needs people” “people with intellectual disabilit*” “people with dementia” “people with cognitive disorder*”	Knowledge Attitude* Perception* Belie* View* Feeling* Experience* Literacy	
**Objective 2**			
Education related	Carers related	Oral health related	People with disability related
Education training program curriculum	carer* caregiver* “family member*” relative* “informal carer*” “paid carer*” “unpaid carer*”	“oral health” “oral hygiene” “dental health” “dental care” “oral care” “mouth care”	“people with disabilit*” “disabled people” “special needs people” “people with intellectual disabilit*” “people with dementia” “people with cognitive disorder*”

**Table 2 healthcare-10-02061-t002:** Overview of studies selected.

Author, Year, Country	Title	Study Type and Category	Study Population	Sample Size	Key Findings
**Objective 1: Education/training-related interventions for dental health professionals**					
Blaizot et al., 2018 France	Obstacles to comprehensive dental care in patients with sustained limitations of their decision-making abilities: findings from a Delphi study	Quantitative/ qualitative/ exploratory/ knowledge, skills, and attitude	Dental practitioners	72	Using the Delphi technique with a panel of 72 dental practitioners, the researchers explored obstacles to comprehensive dental care for people with limited decision-making abilities. Agreement was reached that oral health care requires a high degree of co-operation in care and in hygiene. Obstacles include difficulties in establishing and maintaining long-term oral hygiene behaviours, low caregivers’ priority for oral health, and difficulties in follow-up care. The undergraduate and postgraduate curriculum developed by the International Association of Disability and Oral Health was recommended to enhance a holistic patient-centred approach. Training for dentists, information and training for caregivers, and oral health screening in nursing homes were recommended. Cultural change including oral health specialists, general practitioners, dental organisations, patients, and caregivers was sought.
Ahmad et al., 2014 Malaysia	Special Needs Dentistry: perception, attitudes, and educational experience of Malaysian dental students	Quantitative/ exploratory/ knowledge, skills, and attitude	Final year dental students of 6 universities	289	Questionnaire results revealed 31.7% of dental students were comfortable treating patients with intellectual disability; 50.5% of responding students felt training in SND was inadequate; 89.6% felt clinical training was needed at undergraduate level; 61.2% of students, the majority of whom were women, were interested in postgraduate SND training; 96.9% of respondents felt they needed didactic training on the topic. Earlier exposure to SND training, liaison with external organisations and different clinical settings, support to gain specialist qualifications, and continuous professional development were recommended by the researchers.
Catteau et al., 2013 France	Using e-learning to train dentists in the development of standardised oral health promotion interventions for persons with disability	Quantitative/ experimental	Dental health professionals	24	Dentists completed a 10-module online training course and implemented a standardised intervention in a disability institution. Results of pre- and post-questionnaires used to measure self-efficacy and course satisfaction indicate the online training contributed to positive outcomes.
Dellavia et al., 2009 Italy	Special care dentistry for people with intellectual disability in dental education: an Italian experience	Quantitative/ exploratory/ knowledge, skills, and attitude	Final-year dental students and Deans of Schools	248 students and 4 Deans of Universities	Surveys distributed to 20 Italian dental schools received a 51% student response rate but only 20% response rate from Deans. Surveys sought data on curriculum content, satisfaction with curriculum, and attitudes of students toward dental care for people with intellectual disability. Responses suggest low levels of didactic and clinical training and poor training content. Revised curricular standards were recommended to improve training in treatment of people with intellectual disability.
Faulks et al., 2018 France	Development of a battery of tests to measure attitudes and intended behaviours of dental students towards people with disability or those in marginalised groups	Quantitative/scales or tools to assess knowledge, skills, and attitudes.	Dental health students, graduates, and practitioners who have studied in 4 countries	130	The researchers developed a survey tool and test battery to evaluate values, attitudes, and intentions of dental students towards people with disability. Three adapted scales and the theory of planned behaviour were piloted in 4 countries with students and dental health practitioners at 5 different professional stages. Scales were assessed for internal reliability, variance, distribution and face, and content validation. The tool will be validated for use by educators to evaluate the impact of clinical exposure and teaching on their students
Isobe et al., 2018 Japan	Attitudes towards people with dementia: a cross-sectional study comparing dental hygiene students with registered dental hygienists	Quantitative/ exploratory/ knowledge, skills, and attitude	Registered dental hygienists and dental hygiene students	255	Surveys were used to assess attitudes and knowledge. Likert scale analysis and the Fraboni Ageism Scale found registered dental hygienists had more knowledge and positive attitudes towards people with dementia compared with dental hygiene students. Training in treatment of people with dementia should be added to dental hygienists’ curriculum.
Mac Giolla Phadraig et al., 2015 Ireland	Does special care dentistry undergraduate teaching improve dental student attitudes towards people with disabilities?	Quantitative/ experimental	Third-year dental health students	109	The researchers used pre- and post-testing to assess the impact of a brief blended learning module on special care dentistry over 3 years. No statistically significant difference was found in student attitudes toward people with disability before and after educational intervention. Dental students in this study had neither particularly positive nor negative attitudes towards people with disability. Researchers proposed the scale used to measure attitudinal change was inadequate for the purpose.
Mohebbi et al., 2014 Iran	Effectiveness of training senior dental students on oral health care for disabled patients	Quantitative/ experimental	Dental health students	70 (Control group = 34, experimental group = 36)	Likert scale analysis of pre- and post-surveys of final year dental students about a 10-day intervention comprising presentations, peer discussion, visit to childcare centre for autistic children, and discussion about communication with patients with disability. Results indicated improvements in knowledge and attitudes of the intervention group from short-term training programmes, but researchers recommended further longitudinal research on retention of knowledge and attitudes.
Jones et al., 2018 U.S.A.	Effectiveness of an educational module on dental hygiene students’ attitudes towards persons with disabilities	Quantitative	Dental health students	Pre-training survey = 81 Post-training survey = 65	Questionnaires measured effectiveness of DVD educational module developed inclusively with people with disability and used with 2nd- or 3rd-year dental students. A range of scales was used to measure attitudes and empathy toward people with disabilities pre and post. Results indicated the module was an effective model for patient-centred training. Researchers recommended further qualitative study and longitudinal research into the effect of the module.
Nagarajappa et al., 2012 India	Dentists’ attitude to provision of care for people with learning disabilities in Udaipur, India	Quantitative/ exploratory/ knowledge, skills, and attitudes	Private dental health professionals, faculty, and postgraduate students across Udalpur, India	247	Analysis of questionnaires using Statistical Package for Social Sciences (SPSS) Version 16 found the overall attitude of respondents toward treating people with learning difficulties was positive, with a statistically significant correlation between the mean attitude of dentists and their previous experience of treating people with learning difficulties.
Smith et al., 2010 Ireland	Provision of dental care for special care patients: the view of Irish dentists in the Republic of Ireland	Quantitative/ exploratory/ knowledge, skills, and attitude	Dental health professionals	236	This survey received only a 35% response rate. Analysis using SPSS found 41% of respondents had trained in SND, and 65% expressed willingness to train; 66% had experience in treating people with disability; 59% of respondents who had received no training in treating people with disability were doing so. The. most common service provided in SND was management of dental emergencies. Only 25% of respondents were aware of legal obligations to people with disability. There was emphasis on a need for clinical training at undergraduate and postgraduate level and education and training for carers and community-based preventative care. Behaviour and communication were perceived to be the main obstacles to the provision of dental care.
**Objectives 1 & 4**					
Mac Giolla Phadraig et al., 2016 Ireland	Should we educate care staff to improve the oral health and oral hygiene of people with intellectual disability in residential care? Real world lessons from a randomized controlled trial	Quantitative/ experimental	Carers and adults with intellectual disabilities in residential care	76 (Intervention group = 35, control group = 41)	Researchers trained 22 residential care nurses in oral hygiene, who then trained peers in residential care. Pre- and post-data collection and dental examination of residents by experienced dentist were followed by analysis of clinical and descriptive data and comparison between intervention and control groups. Results suggested the intervention did not lead to improved oral hygiene, but there were improvements in knowledge, attitudes, and beliefs of carers about oral hygiene of people with intellectual disability.
**Objective 3: Education/training-related interventions for people with intellectual disabilities**					
Lees et al., 2016 U.K.	Adults with learning disabilities experience of using community dental services: Service user and carer perspectives	Qualitative/ exploratory	Carers and adults with intellectual disabilities	4 adults with learning difficulties and 6 carers	Participant interviews and thematic analysis found people with intellectual disability expected dental staff to understand their needs. Positive factors were effective interpersonal communication between dental staff and participants, providing information, relieving anxiety, actively listening, and showing dignity. Setting aside enough time for explaining procedures during consultations enhanced the experience patients. Further studies are required with a wider population of people with intellectual disability who use dentists.
Mac Giolla Phadraig et al., 2016 Ireland	What should dental services for people with disabilities in Ireland be like? Agreed priorities from a focus group of people with learning disabilities	Qualitative/ experimental	People with intellectual disabilities	6	The researchers conducted a single focus group with six adults with learning difficulties and cognitive/communication disorders, using accessible methods. Priorities identified by people with learning disabilities were information, treatment, access, quality of care, and cost. The researchers identified disempowerment of people with learning difficulties and proposed dental services could improve the accessibility and content of information for this group. The researchers identified the need to train dental health professionals in communication with people with learning disabilities and argued that people with learning disabilities should be involved in planning and evaluating oral health services.
**Objective 4: Education/training-related interventions for carers of people with intellectual disability**					
Pradhan et al., 2016 Australia	Pre- and post-training evaluation of dental efficacy and activation measures in carers of adults with disabilities in South Australia—A pilot study	Quantitative/ experimental	Carers	41 carers of 103 adults with disabilities	Carers were trained by a SND in oral health care, supported by dental hygienist for 2 months and dental visit at 3 months. Pre- and post-questionnaires on dental behaviours of carers and carers’ knowledge, skill, confidence, and dental efficacy. Results indicated increased knowledge and confidence but not skills. Carers were better able to assess the referral needs of their care recipients’ oral health similar to the dentist, indicating combined theoretical and practical training and continued support can improve the knowledge and confidence of non-dental health professionals such as carers in providing oral care for their recipients. Further research in a larger study is needed.
Phlypo et al., 2018 Belgium	Integrating community service learning in undergraduate dental education: A controlled trial in a residential facility for people with intellectual disabilities	Quantitative	Carers of adults with intellectual disability in residential care	37 = Residents (Intervention group = 18); 55 = Caregivers (Intervention group = 24)	An intervention in a residential care service for people with intellectual disability comprised information for caregivers, pre-and post-test dental examinations of residents, and a questionnaire assessing caregivers’ knowledge. Compared with the control group, the gingival health of residents after the test was found to have improved. Overall, there was some but limited impact on caregivers’ knowledge and residents’ oral health. Researchers made recommendations on future research in community service learning.
Mac Giolla Phadraig et al., 2012 Ireland	Train the trainer? A randomized controlled trial of a multi-tiered oral health education programme in community-based residential services for adults with intellectual disability	Quantitative/ experimental	Carers	Intervention group =21, control group =29	21 residential care staff received multi-tiered, day-long training from dental health professionals on oral health, oral disease, and prevention of oral disease. Topics were brushing, diet, dealing with challenging behaviours, and care of complete dentures. A control group received no training. Pre- and post-tests of knowledge, attitude, self-efficacy, and self-reported behaviour indicated some improvement in knowledge but no statistical improvement in attitudes. Researchers partially attributed results to increased awareness on the part of the control group (Hawthorne effect) and limitations in research methods but concluded that multi-tiered training programs can improve oral-health-related knowledge, attitude, self-efficacy, and reported behaviour in residential care settings for people with intellectual disability.
Mac Giolla Phadraig et al., 2015 Ireland	Should we provide oral health training for staff caring for people with intellectual disabilities in community based residential care? A cost-effectiveness analysis	Cost effectiveness evaluation/experimental	N/A	N/A	Analysis of capital and operating expenses in conducting the above train-the-trainer intervention, despite its small effect, found it to be cost-effective in improving oral-health-related knowledge and reported behaviour, attitudes, and self-efficacy. Ongoing research into cost benefit is required.
Eijsink et al., 2018 Netherlands	A Q-methodology study among caregivers of people with moderate intellectual disabilities on their clients’ health care: An example in oral health	Qualitative/ quantitative exploratory	Carers of people with moderate intellectual disabilities living in institutions	40	The researchers used Q-methodology to explore carers’ viewpoints about oral hygiene. By-person factor analysis was used to cluster caregivers’ views. Four factors were identified: 1. Responsible and perseverant; 2. Motivated and aware of obstacles; 3. Social-minded and knowledgeable; 4. Concerned and insecure. Researchers concluded that results can be used to determine attitudes of caregivers regarding oral healthcare and to consider dental care professionals’ adaptation of advice to carers’ specific characteristics.
Rozas et al., 2017	Strategies to improve dental health in elderly patients with cognitive impairment	Systematic review	Elderly people with cognitive impairment or dementia and carers	9 full-text articles	Basic oral care plans for people with cognitive impairment should, at minimum, focus on preventative strategies recommended for healthy older patients. Oral hygiene education should be promoted by dental health professionals for caregivers of people with cognitive impairment.

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
