# Peer review of "Education/Training Interventions to Improve Oral Health of Adults with Intellectual Disability"

_healthcare, 2022, doi:10.3390/healthcare10102061_

Round 1
Reviewer 1 Report
Authors present what they describe as a scoping review that explored education/training interventions related to the oral health of adults with intellectual disabilities. Persons with disabilities are at high risk for poor oral health and systemic consequences of poor oral health. They hoped to study interventions targeting dental health professionals, non-dental health professionals, adults with intellectual disabilities, and their paid and unpaid supporters. Six electronic databases were searched using PRISMA guidelines and Crowes Critical Appraisal Tool.
Scoping reviews are an increasingly popular methodology that promote knowledge synthesis of abundant volumes of primary research, such as systematic reviews.{Arksey, 2005;Pham, 2014;Colquhoun, 2014} In scoping reviews, authors must provide a rationale for the themes they choose to select/review for knowledge synthesis. The authors have done this in describing their four objectives. However in the Methods section, authors do not differentiate this study as a scoping review as described by the references above. They only state that they used PRISMA guidelines which are defining for systematic reviews. Therefore, I suggest that they provide a few sentences after their PRISMA statement as evidence that this study is actually a scoping review with thematic knowledge synthesis objectives, that they provide in their rationale at the end of the Introduction. This will then easily connect with the inclusion and exclusion criteria that follow and provide a better flow. The section on study selection thereafter is excellent and fitting to the concept of a scoping review.
The study, in general, is well thought out, executed and informative. So only this minor revision required.
Reviewer 2 Report
Dear authors
I carefully read the article entitled " Education/ training interventions to improve oral health of 2adults with intellectual disability". I found the work interesting to read. However, I think there is a part in the paper that needs to be revised. Please see below the comments I would like to see addressed.
Abstract
Line16: Please briefly add the criteria for the paper to be selected after the sentence, 'It is a qualification for inclusion of 20 papers.'
Introduction
1. Line 57: in my countries : Please write the specific name of the country.
2. Line 58: What are the main services of public dental services in Australia? You need a brief description.
.
Methods
3. Dental health professionals and non-dental health professionals Scope of occupation? Operational definition is required.
4. Please add a footnote for the meaning of ‘*’ in Table 1
5. Line 152: this data was verified by the other two investigators.
Is there an academic standard for the number of researchers when verifying data? If you do, please add it to the research method
Discussion
6. Line: 244~255:Isobe, Izumi [21] similarly found attitudes of dental hygienists toward adults with dementia were more positive than those of dental hygiene students [21].
In response, please add to the consideration of what changes should be required in students' education.

Round 2
Reviewer 2 Report
I have no other opinion. thank you!